# Transarterial Embolization for Spontaneous Soft-Tissue Hematomas: Predictive Factors for Early Death

**DOI:** 10.3390/jpm13010015

**Published:** 2022-12-22

**Authors:** Rémi Grange, Lucile Grange, Clément Chevalier, Alexandre Mayaud, Loïc Villeneuve, Claire Boutet, Sylvain Grange

**Affiliations:** 1Department of Radiology, University Hospital of Saint-Etienne, 42270 Saint-Priest-en-Jarez, France; 2Department of Internal Medicine, University Hospital of Saint-Etienne, 42270 Saint-Priest-en-Jarez, France

**Keywords:** hematoma, soft-tissue, embolization, anticoagulant, bleeding

## Abstract

Introduction: The aim of this retrospective monocentric study was to assess the safety and efficacy of spontaneous soft-tissue hematoma transarterial embolization (TAE) and to evaluate predictive factors for early mortality (≤30 days) after TAE for spontaneous soft-tissue hematoma (SSTH). Materials and methods: Between January 2010 and March 2022, all patients referred to our hospital for spontaneous soft-tissue hematoma and treated by emergency TAE were reviewed. Inclusion criteria were patients: ≥18-year-old, with active bleeding shown on preoperative multidetector row computed tomography, with spontaneous soft-tissue hematoma, and treated by TAE. Exclusion criteria were patients with soft-tissue hematomas of traumatic, iatrogenic, or tumoral origin. Clinical, biological, and imaging records were reviewed. Imaging data included delimitation of hematoma volume and presence of fluid level. Univariate and multivariate analyses were performed to check for associations with early mortality. Results: Fifty-six patients were included. Median age was 75.5 [9–83] ([Q1–Q3] years and 23 (41.1%) were males. Fifty-one patients (91.1%) received antiplatelet agent and/or anticoagulant therapy. All 56 patients had active bleeding shown on a preoperative CT scan. Thirty-seven (66.0%) hematomas involved the retroperitoneum. Median hemoglobin level was 7.6 [4.4–8.2] g/dL. Gelatine sponge was used in 32/56 (57.1%) procedures. Clinical success was obtained in 48/56 (85.7%) patients and early mortality occurred in 15/56 (26.8%) patients. In univariate and multivariate analysis, retroperitoneal location and volume of hematoma were associated with early mortality. Conclusion: Retroperitoneal location and volume of hematoma seem to be risk factors for early death in the context of TAE for spontaneous soft-tissue hematoma. Larger multicenter studies are necessary to identify others predictive factors for early mortality and to anticipate which patients may benefit from an interventional strategy with TAE.

## 1. Introduction

Spontaneous soft-tissue hematoma (SSTH) is a relatively rare but potentially life-threatening condition [1]. The prevalence of SSTH is expected to increase due to the increasing use of anticoagulants among older patients and its diagnosis due to routine use of multidetector row computed tomography (MDCT) [2]. SSTH may cause pain, deglobulization, hemodynamic instability, and death [3]. SSTH may cause hospitalization or a clinically aggravating event in a hospitalized patient, especially in intensive-care units (ICUs) [4]. Its diagnosis may be delayed in the absence of apparent symptoms. There is no common consensus for appropriate SSTH management. Different options include conservative treatment or transarterial embolization (TAE). Surgical treatment [5] should be reserved as a last resort if TAE is unsuccessful. Moreover, depending on the clinical severity and history of the patient, preservation, discontinuation, or reversal of anticoagulant treatment should be discussed with the management team [6]. Preoperative MDCT plays a central role in locating the hematoma by assessing its volume and muscular relationships and by anticipating emergency TAE [7]. Nevertheless, the injection of contrast media may aggravate acute renal failure and require extracorporeal purification [8,9]. On the one hand, TAE is effective in stopping SSTH bleeding [10,11] but may generate or aggravate acute renal failure of multi-factorial origin in the aftermath of the acute bleed. On the other hand, active bleeding may stop spontaneously, especially in the absence of fascial rupture by intrinsic muscle compression. The clinical success rate of TAE for SSTH is estimated at 94.3% [3]. However, the mortality rate of these patients is high in 22.7% of the cases, with an early rebleeding rate of 9.4% [3]. Even if there is no standard of care for the diagnosis and treatment of SSTH, a management algorithm has been proposed [12]. Previous studies have focused on the occurrence of SSTH in patients admitted to intensive care [13], in patients treated with anticoagulants [6,14], and experiencing SARS-CoV-2 infection [15]. Moreover, studies on TAE in SSTH focused on a particular muscle [16] or a particular embolic agent [17,18]. Evaluating predictive factors for early mortality after TAE of SSTH has received little research attention: Barral et al. [7] showed that hematoma volume, retroperitoneal hematoma, and simplified acute physiology score II were independent predictors of early mortality after TAE. It is, therefore, essential to better understand the predictive factors of mortality to improve the selection of patients who may benefit from emergency TAE.

This retrospective monocentric study aimed to assess the safety and efficacy of spontaneous soft-tissue hematoma transarterial embolization and to evaluate predictive factors for early mortality after TAE for SSTH.

## 2. Methods

### 2.1. Study Population

Between January 2010 and March 2022, we reviewed clinical decisions and MDCT images for all patients referred to our hospital for SSTH treated by emergency TAE. The inclusion criteria were: patients (1) ≥18 years old, (2) with active bleeding shown on MDCT, (3) and treated by TAE. The exclusion criteria were: patients (1) <18 years old, (2) with soft-tissue hematomas of traumatic, (3) iatrogenic, or (4) tumoral origin, and (5) without preoperative MDCT angiography before TAE.

### 2.2. Patient Characteristics

We retrospectively reviewed medical records for old clinical history, including high blood pressure, diabetes mellitus, chronic renal failure, cardiovascular diseases, cirrhosis, performance status, antiaggregant and anticoagulant treatments, and indications to administer anticoagulant therapy. The recent clinical history included the presence and reasons for hospitalization at the time of diagnosis. Biological variables were collected at presentation, before TAE, including serum hemoglobin level, prothrombin time (PT), international normalized ratio (INR), and platelet count. The numbers of red blood cells, fresh frozen plasma, and platelets transfused were also reported. We considered transfusions performed within 24 h before and 48 h after the procedure. Hemoglobin drop was defined as the difference between the baseline hemoglobin level and hemoglobin level at the time of the procedure. Hemodynamic instability was defined as a decrease in blood pressure requiring the use of amines. Discontinuation, reversion of anticoagulant therapy, and the type of medication used to achieve reversion were reviewed. 

### 2.3. Imaging and Procedure Data

All patients underwent an abdominal MDCT scan (SOMATOM SENSATION before September 2014 and SOMATOM, Siemens^®^ AG, Medical solutions, Erlangen, Germany). Patients received ≥90 mL contrast medium (Xenetix 350, Guerbet^®^, Villepinte, France) with a flow rate ≥ 3 mL/s. Acquisitions without and with the injection of iodinated contrast medium at the arterial and portal phase were routinely performed in order to demonstrate active bleeding, defined by the contrast medium leaking during the arterial phase in MDCT images that grows during the portal phase. The location of and number of hematomas were recorded. Hematoma volume was evaluated by semi-automatic delimitation using the Carestream^®^ software (Rochestern, NY, USA) by one experienced radiologist. In the case of multiple locations, the sum of the volume of all hematomas was calculated. The presence of a fluid level within the hematoma was also assessed.

All emergency TAEs were performed by 5 interventional radiologists with at least 3 years of experience in performing TAEs, after a collegial discussion between the interventional radiologist, the intensive-care physician, and the surgeon. After local anesthesia with 5% lidocaine, the right common femoral artery was accessed. First, global aortography was performed with a Pigtail 5F probe and a hydrophilic guidewire (Terumo, Tokyo, Japan). Then, a catheterization of the artery feeding the bleeding was performed with a Cobra 5F probe, and a 2.7F supraselective microcatheteter (Progreat, Terumo^®^, Tokyo, Japan) was used at the discretion of the interventional radiologist. TAEs were performed under fluoroscopic monitoring using micro coils (Interlock and IDC, Boston^®^ Scientifics), N-butyl-2-cyanoacrylate (Glubran^®^ GEM, Viareggio, Italy), gelatine sponge (Gelitaspon^®^), or microparticles (Embosphere^®^, Microspheres, BioSphere Medical, Rockland, MA, USA), depending on the location of the active bleeding, the intensity of the active bleeding, the presence of collateral arteries, the clinical severity, and the habits of the interventional radiologist. In the absence of active bleeding, an empirical TAE could be performed in case of hemodynamic instability based on the data from the preoperative MDCT. Complete fluoroscopic angiography controls were performed in order to confirm that bleeding had been successfully controlled. After the treatment, the introducer was sutured to the skin and removed the following day if there was no recurrence of bleeding. After TAE, all patients were monitored closely in the ICU for clinical signs and symptoms that were potentially suggestive of ischemic complication or recurrent bleeding until discharge or death.

### 2.4. Outcomes

These clinical findings were supplemented by laboratory studies. The long-term outcomes of the patients, specifically the incidence of rebleeding, mortality, and procedure-related complications, were determined by chart review. MDCT angiography following TAE was not routine practice in the hospital unit during this period. Technical success was defined as the stopping of active bleeding, based on the angiographic control findings. Clinical failure was defined as bleeding-related death or rebleeding that required repeat TAE during the 30-day follow-up period. In the case of suspected recurrent bleeding, MDCT with contrast injection was performed. Recurrent bleeding was defined by the presence of active bleeding on the follow-up MDCT after the TAE. Rebleeding was defined as early if it occurred within 30 days of TAE and as a late rebleeding event if it occurred after 30 days of TAE. Overall survival was calculated from the date of TAE until death from any cause. Complications of TAE were defined as ischemic complication in embolized territories, non-target embolization, or hematoma at the puncture site. Complications were defined as early when they occurred within the first 2 h following TAE and late complications at least 24 h after the procedure. Grades A and B were considered to be minor complications and grades C, D, E, and F were considered to be major complications according to the SIR classification [19].

### 2.5. Statistical Analysis

Data are presented either as absolute numbers with percentages for qualitative variables or as median (Q1–Q3) for quantitative variables. Univariable and multivariate analyses were conducted by using the Cox proportional hazard model to identify potential prognostic factors of survival and to estimate the adjusted odds ratio (OR) with 95% CI. R^®^ software 3.6.2(R Foundation for Statistical Computing, Vienna, Austria) was used for this study. A Kaplan–Meier curve was created with Prism Graphpad^®^ software 8.4.2 (GraphPad Software Inc., San Diego, CA, USA).

### 2.6. Ethical Considerations

This study was performed in accordance with the ethical standards of the Helsinki Declaration and was approved by the ethic committee of the University Hospital of Saint-Etienne.

## 3. Results

Between January 2010 and March 2022, 63 patients were referred to TAE for SSTH in our institute. Seven patients had angiography without TAE: six patients had negative angiography without TAE and one had a technical failure, resulting in a total of 56 patients treated with TAE.

### 3.1. Patient Characteristics

Patient characteristics are detailed in Table 1. Our study population included 56 patients. The median age was 75.5 (39–83) years and 23/56 (41.1%) were men. Among the population, 24/56 (42.8%) patients had a Performance Status ≥ 2. Regarding comorbidities, 27/56 (48.2%) patients had high blood pressure, 16/56 (33.9%) patients had diabetes, 32/56 (57.1%) patients had cardiovascular disease, 9/56 (16.1%) patients had chronic renal failure, 12/56 (21.4%) patients had a history of cancer, and 4/56 (7.1%) had cirrhosis. Before the occurrence of SSTH, 24/56 (42.9%) patients were already hospitalized. In 26/56 (46.4%) patients, no clinical symptoms except hypotension and/or blood loss were found. Of the 56, 5 (8.9%) patients received antiplatelet therapy, 39/56 (69.6%) patients received anticoagulation therapy, 7/56 (12.5%) received both antiplatelet and anticoagulation therapy, and 5/56 (8.9%) did not receive antiplatelet or anticoagulation therapy. The two main indications for antithrombotic therapy were cardiac arrhythmia in 27/51 (53.0%) patients and deep vein thrombosis in 10/51 (19.6%). Overdosage was found in 13/51 (25.5%) patients. All patients reported the discontinuation of antithrombotic therapy and 13/25 (25.5%) patients reported the reversion of anticoagulation. A clinical symptom was present in 30/56 (53.6%) patients, including abdominal pain (n = 26) and tumefaction (n = 4).

Concerning biological data, the median Hb was 7.6 (4.4–8.2) g/dL, the median drop of Hb was 3.45 (0.9–4.9) g/dL, and 41/56 (73.2%) patients received red blood cell (RBC) transfusion. The median INR was 1.45 (1–2.7) the median PT was 65 (8–75)% and the median platelet count was 180 (38–255) G/L. A total of 41 (71.9%) patients had red blood cell transfusion and the median red blood cell units transfused was 4 (1–7.5) per patients. A total of 25 (44.6%) patients had fresh frozen plasma and 7/56 (12.5%) patients had platelet transfusion. The median fresh frozen plasma and platelet units transfused was 3 (1–6) and 2 [2,3] per patient, respectively. Biological data are detailed in Table 2.

### 3.2. Imaging and Procedure Data

On MDCT, the hematoma was located in the retroperitoneum in 37/56 (66.1%) patients, the rectus sheath in 13/56 (23.2%) patients, and the thigh muscle in 3/56 (5.4%) patients. Three patients (3.2%) had two SSTHs: one patient had psoas and thigh hematomas, one patient had rectus and thigh hematomas, and one patient had thigh and retroperitoneal hematomas. The median volume of SSTH was 1336 (135–1664) ml and 32/56 (57.1%) patients had a fluid level within the hematoma(s). Pre-procedure data are detailed in Table 2.

Angiographic active bleeding was detected in 50/56 (89.3%) patients, resulting in 6/56 (10.7%) patients being treated by empiric TAE. Procedures were performed using a gelatine sponge for 32/56 (57.2%) patients, microparticles for 11/56 (19.6%) patients; N-Butyl Cyanoacrylate (NBCA) for 4/56 (7.1%), with a combination of sponge and microparticles for 2/56 (3.6%) patients; a combination of gelatine sponge and NBCA in 1/56 (1.8%) patients; and a combination of coils, gelatine sponge, and particles for 1/56 (1.8%) patients (Figure 1).

The main arteries embolized were: the lumbar artery in 30/56 (53.6%) patients, the inferior epigastric artery in 15/56 (26.8%) patients, and the ilio-lumbar artery in 14/56 (25.0%) patients. Multiple arteries were embolized in 8/56 (14.3%) patients. Two hematomas at the puncture site were reported (Grade A). Pre-procedure data are detailed in Table 3.

### 3.3. Outcomes and Prognostic Factors

Within 30 days, 8/56 (14.3%) patients had a recurrence of bleeding, all of whom were treated by repeat TAE. A flowchart of the patient outcomes is illustrated in Figure 2. One patient had a third TAE. During the follow-up period, 20/56 (35.7%) patients died. Of these, 15/56 (26.8%) deaths occurred ≤30 days and 5/56 (8.9%) deaths occurred >30 days after TAE (Figure 3). Of the patients who had a bleeding recurrence, 4/8 (50%) died ≤30 days. No surgical management or radiological drainage of the hematoma was performed. One patient with a retroperitoneal hematoma had partial ischemia of the kidney on arterial plication by extrinsic compression of the hematoma. The other patients did not present symptomatic compression of the surrounding organs. No related TAE-delayed complications were noticed. Figure 4 shows a Kaplan–Meier survival curve of the study population. Patient outcomes are detailed in Table 4.

Univariate and multivariate analyses are detailed in Table 5. There was no association between early death and age, sex, performance status, target artery, or the use of coils.

In univariate analysis, the retroperitoneal location (OR = 4.65 (1.32–13.31), *p* = 0.016) and the volume of hematoma (OR = 7.70 (1.86–31.92), *p* = 0.004) were associated with early mortality.

In multivariate analysis, the retroperitoneal location (OR = 4.08 (1.01–61.4), *p* = 0.047) and the volume of hematoma (OR = 1.01 (1.00–1.02), *p* = 0.023) were associated with early mortality.

## 4. Discussion

This retrospective study showed that emergency TAE of SSTH allows for the rapid control of bleeding, with high clinical success (85.7%), a relatively high early mortality rate (26.8%), and a significant rebleeding rate (14.3%). Moreover, it demonstrated an association between the hematoma volume, retroperitoneal location, and overall survival in multivariate analysis.

There are no official recommendations for managing patients treated with SSTH, especially those patients receiving thrombotic therapy. A review by Touma et al. [3] reported a high clinical success (93.1%), a moderate overall mortality rate (22.8%), and a significant recurrence rate (10.1%). This review showed that only 10 studies included >4 patients.

In line with previous studies, the present one showed a significant rate of early mortality rate (26.8%) within the first 30 days following TAE in patients treated for SSTH. That said, TAE was able to stop bleeding in all procedures. SSTH appears as a triggering event or may aggravate a precarious clinical situation in polypathological patients. Barral et al. [7] reported a 27% 30-day mortality rate following TAE for SSTH, despite high clinical success (83%). Our study found a gap between significant technical success (98.2%), clinical success (85.7%), and early mortality rate (26.8%). It is, therefore, essential to adopt an aggressive strategy in patients experiencing recurrences of bleeding and to monitor them closely for early recurrence.

We found that the retroperitoneal location and hematoma volume were associated with early mortality following TAE. Barral et al. [7] also found that retroperitoneal hematoma location was associated with mortality within 30 days. This can be explained by the higher rate of fascia rupture, resulting in a larger hematoma volume and a delayed symptomatology, leading to delayed medical management [7]. Volume measurement can be performed semi-automatically or with three measures within seconds on the pre-therapeutic CT scan and is a simple, quick way to argue for a therapeutic decision. The fluid level was not associated with early mortality in our study. Nakayama et al. [20] showed fluid level in 28/47 (60%) MDCT of patients with coagulopathy-related SSTH and an association between fluid level and active bleeding. However, the fluid level was only present in 32/56 (57.1%) patients with active bleeding in this study and was not associated with early mortality after TAE. Therefore, the relevance of its description in patients receiving preoperative MDCT angiography seems modest.

Almost all patients included in the study had anticoagulant and/or antiplatelet therapies at the time of TAE. This shows the important association between antithrombotic use and the occurrence of SSTH, which has already received attention in the published literature [3]. Barral et al. [7] and Popov et al. [12] recommended conservative treatment in patients with spontaneous soft-tissue bleeding, with active bleeding on CT and/or hemodynamic instability, and without fascia rupture. However, the withdrawal and reversal of anticoagulants are subject to the benefit–risk balance in patients with cardiovascular co-morbidities, particularly patients with cardiac rhythm disorders at risk of embolism. In our study, the reversal of anticoagulant is moderate (25.5%), highlighting the apprehension of reversing anticoagulants in patients with thrombogenic risk.

The present study showed a moderate rate of 14.3% of recurrent bleeding after TAE, which is in line with previous literature. Dohan et al. [21] showed rebleeding in 9/34 (26.4%) patients. This can be explained by the continued use of anticoagulants or by the delay in the reversion of anticoagulants. It is worth noting that all eight patients who presented with recurrent bleeding had clinical success during a repeat TAE. However, the early mortality rate in these patients is relatively high (50%), as was the case in Dohan et al.’s study [21], which showed 4/9 (44.4%) patients who experienced early death after presenting with a recurrence of bleeding after TAE.

MDCT angiography showed excellent sensitivity for detecting active bleeding of SSTH. There is no consensus on the pertinence of preoperative MDCT, which is not routinely performed. Vincenzo et al. [6] performed a systematic angiography in patients with persistent bleeding requiring iterative transfusions but not necessarily preoperative MDCT, which explains the large number of negative angiograms (25%) in their study. MDCT can limit the injection of contrast medium during TAE, to limit irradiation and to accelerate the TAE procedure by directly catheterizing the target artery. Contrary to Barral et al. [7], an MDCT angiogram was performed before TAE for all patients in the present study. However, we reserved iodine contrast injection for patients with hemodynamic instability and a glomerular filtration rate <30 mL/min or for patients with a glomerular filtration rate ≥30 mL/min in cases with a suspicion of active bleeding. The preoperative MDCT scan has a fundamental role in the choice of the target artery. In addition, it can avoid unnecessary angiography in patients without active bleeding. In fact, the sensitivity of MDCT is more effective at detecting active bleeding than angiography (≥0.3 vs. 0.5 mL/min, respectively) [22,23]. The benefit–risk balance must be assessed, without underestimating the expected benefits of the injection of iodine contrast on patient management, especially since iodine-induced nephropathy has often been overestimated in studies [8].

Complications following TAE of SSTH are rare. Our study showed two puncture-site hematomas, without any ischemic muscle complications or worsening of pain that might suggest a muscle infarction, in line with the literature. However, cases of parietal muscle necrosis have been described with NBCA [24].

In our study, using a non-absorbable agent was not associated with a worse clinical prognosis than non-absorbable agents. This is in line with the literature, which does not find any influence of the embolizing agent on the prognosis [3]. However, given the low rate of ischemic complications [3] and the high risk of recurrence, a permanent embolizing agent should be prioritized to avoid any recurrence of bleeding on the embolized artery, in case the target artery can be selectively catheterized.

The present study has some limitations. It is a single-center retrospective study, with a limited number of patients; therefore, the results must be interpreted cautiously. All patients had pre-procedural MDCT, which does not reflect the habits of other hospitals and may lead to a selection bias. We studied only patients treated with TAE. The severity of the patients in this population does not reflect the population of patients followed for SSTH.

In conclusion, our study confirmed the safety and efficacy of TAE for SSTH. Nevertheless, early death remains high. The retroperitoneal location and hematoma volume seem to be risk factors for early death. Future multicenter studies are necessary to stratify the risk of early mortality and to anticipate which patients may benefit from an interventional strategy with TAE. 

## Figures and Tables

**Figure 1 jpm-13-00015-f001:**
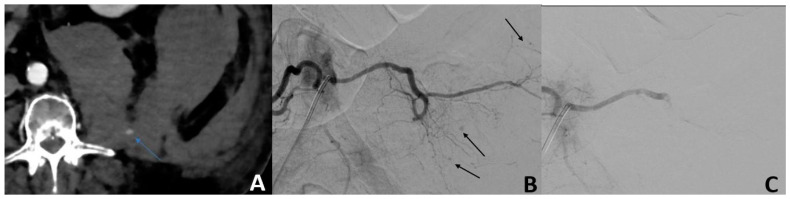
Isolated blood loss in a 74-year-old patient 2 days after cardiac surgery. (**A**) The MDCT scan showed active bleeding (blue arrow) from a large left retroperitoneal hematoma. (**B**) Angiography confirmed active multifocal bleeding from the left L5 lumbar artery (black arrows). (**C**) After embolization with NBCA, the control showed no opacification of the distal branches of the left L5 lumbar artery.

**Figure 2 jpm-13-00015-f002:**
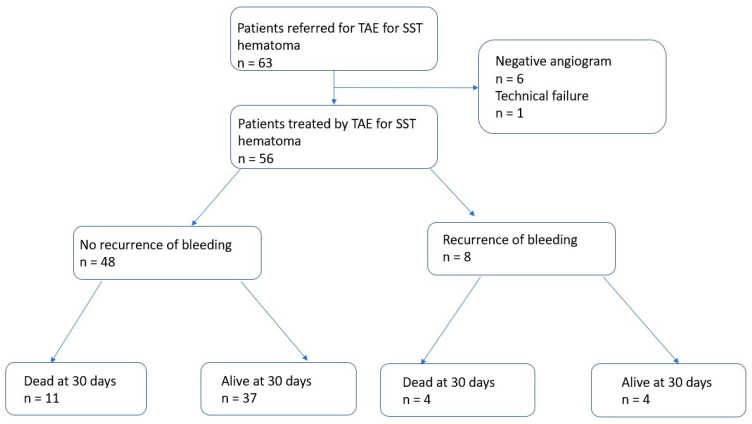
Flowchart of patient outcomes.

**Figure 3 jpm-13-00015-f003:**
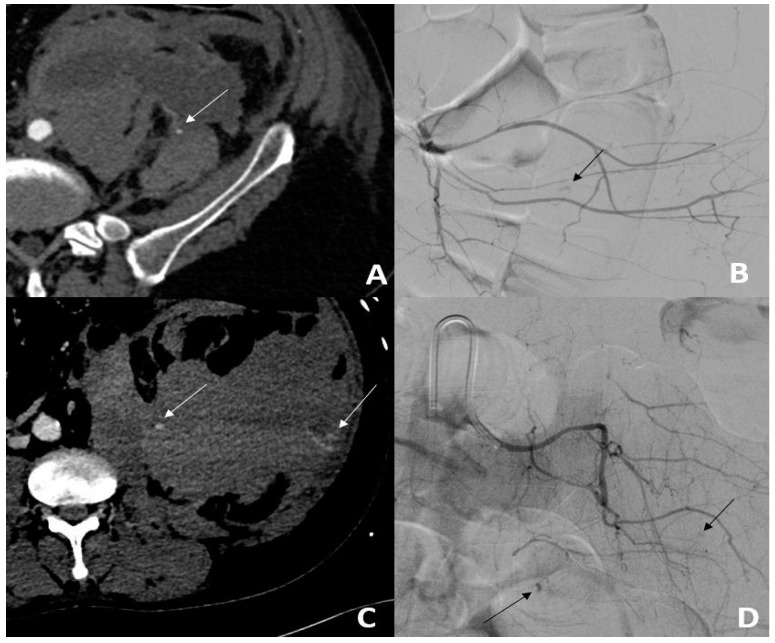
Back pain in a 45-year-old patient hospitalized for Sars-cov2 infection and pulmonary embolism. (**A**) The MDCT scan showed active bleeding (white arrow) from a large left retroperitoneal hematoma. (**B**) Angiography confirmed active unifocal bleeding (black arrow) from the left L5 lumbar artery, treated with gelatine sponge (**C**). On day 1, the patient presented with persistent blood loss. The MDCT scan showed a persistent hematoma with active bleeding (white arrows). (**D**) Angiography showed new multifocal active bleeding (black arrows) treated using 1 coil.

**Figure 4 jpm-13-00015-f004:**
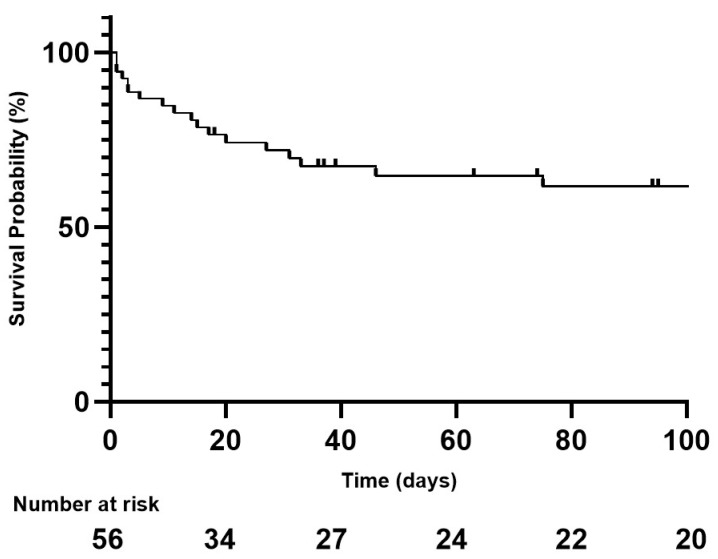
Kaplan–Meier curve of overall survival during the first 100 days following TAE.

**Table 1 jpm-13-00015-t001:** Patient characteristics.

Variables	56
**Age, years** median [Q25–75]	75.5 [39–83]
**Male n. (%)**	23 (41.1)
**Performance Status** median [Q25–75]	1 [0–2]
**Comorbidities n. (%)**	
Diabete	16 (33.9)
HBP	27 (48.2)
Chronic renal failure	9 (16.1)
History of cancer	12 (21.4)
Cirrhosis	4 (7.1)
**Antithrombotic therapy, n. (%)**	**51 (91.1)**
**Indication for antithrombotic therapy, n. (%)**	
Atrial fibrillation	27 (53.0)
Deep venous thrombosis	10 (19.6)
Ischemic stroke	5 (9.8)
Cardiopathy	3 (5.9)
Mechanical valve prosthesis	2 (3.9)
Prophylaxy	2 (3.9)
Others	2 (3.9)
**Hospitalized at the time of diagnosis n. (%)**	24 (42.9)
**Causes of hospitalization n. (%)**	
Infection	7 (12.5)
Surgery	8 (14.3)
Cardiovascular disease	7 (12.5)
Infection and surgery	2 (3.6)
**Hemodynamic instability n. (%)**	30 (53.6)
**Main symptom n. (%)**	
Abdominal pain	26 (46.4)
Tumefaction	4 (7.2)
No clinical symptom	26 (46.4)

**Table 2 jpm-13-00015-t002:** Patient biological data.

Variables	56
**Antithrombotic therapy n. (%)**	**51**
Antiplatelet	**5 (8.9)**
*Clopidogrel*	*2*
*Aspirin*	*3*
Anticoagulant	**39 (69.6)**
*LMWH*	*7*
*UFH*	*14*
*VKA*	*12*
*Apixaban*	*3*
*Rivaroxaban*	*2*
Antiplatelet and Anticoagulation	**7 (12.5)**
*VKA + Aspirin*	*1*
*UFH + Aspirin*	*6*
No antithrombotic therapy	5 (8.9)
**Discontinuation of anticoagulant therapy n. (%)**	51 (100)
**Overdosage n. (%)**	13 (25.5)
**Reversion of anticoagulant therapy n. (%)**	13 (25.5)
Prothrombin complex concentrates	10 (19.6)
K Vitamin	9 (17.6)
Tranexamic Acid	5 (9.8)
**Biology**, median [Q25–75]	
INR	1.45 [1–2.7]
PT(%)	65 [8–75]
Platelet (G/L)	180 [38–255]
Hemoglobin (g/dL)	7.6 [4.4–8.2]
Drop of Hemoblogin (g/dL)	3.45 [0.9–4.9]
**RBC Transfusion n. (%)**	41 (71.9)
Number of RBC, median [Q25–75]	4 [1–7.5]
**Fresh frozen plasma transfusion n. (%)**	25 (44.6)
Number of fresh frozen plasma units, median [Q25–75]	3 [1–6]
**Platelet transfusion n. (%)**	7 (12.5)
Number of platelet units, median [Q25–75]	2 [2,3]

Abbreviations: INR: international normalized ratio, PT: prothrombin time test, UFH: Unfractionated heparin, LMWH: Low-molecular-weight Heparin.

**Table 3 jpm-13-00015-t003:** Pre-procedure patient characteristics.

Variables	56
**Preoperative CT n. (%)**	
Active bleeding	56 (100)
Volume of hematoma median [Q25–75)]	1336 [135–1664]
Fluid level	32 (57.14)
≥2 locations of hematoma	3 (5.4)
**Location of hematoma n. (%)**	
Retroperitoneum	37 (66.0)
Rectus sheath	13 (23.2)
Thigh	3 (5.4)
Psoas + Thigh	1 (1.8)
Rectus sheath + thigh	1 (1.8)
Rectus sheath + retroperitoneum	1 (1.8)
**Angiographic data n. (%)**	
Active bleeding	50 (89.3)
Empirical Embolization	6 (10.7)
**Arteries Embolized n. (%)**	
Lumbar	30 (53.6)
Ilio-lumbar	14 (25.0)
Epigastric inferior	15 (26.8)
Deep femoral	4 (7.1)
**Number of embolized arteries n. (%)**	
*1*	48 (85.7)
≥*2*	8 (14.3)
**Embolic Agents n. (%)**	
Gelatine sponge	32 (57.2)
Microparticles	11 (19.6)
Coils	5 (8.9)
NBCA	4 (7.1)
NBCA + gelatine sponge	1 (1.8)
Microparticle + gelatine sponge	2 (3.6)
Microparticle + gelatine sponge + coils	1 (1.8)
**Time of procedure (min)** median [Q25–75]	43 [16–60]

Abbreviations: NBCA: N-Butyl-Cyanoacrylate.

**Table 4 jpm-13-00015-t004:** Patient outcomes.

Variables	56
Clinical Success n. (%)	48 (85.7)
Mortality during follow-up n. (%)	20 (35.7)
Day-30 mortality n. (%)	15 (26.8)
Day-3 mortality n. (%)	5 (8.9)
Per-operative Complications n. (%)	2 (3.6)
Post-Operative Complications n. (%)	0
Recurrence of Bleeding n. (%)	8 (14.3)
Early ≤ 30 days	8 (14.3)
Delayed > 30 days	0
Management of Early Rebleeding n. (%)	
Repeat TAE	8/8 (100)
Duration of follow-up (days) median [Q25–75]	31 [0–210]

Abbreviations: TAE: Transarterial embolization.

**Table 5 jpm-13-00015-t005:** Univariate and multivariate regression analysis for early death.

		Early Death ≤ 30 Days		
	Univariate Analysis		Multivariate Analysis	
Characteristics	OR	*p* Value	OR	*p* Value
Demographics data				
Age (years)	1.03 (0.98–1.10)	0.23	-	-
Male	1.36 (0.41–4.50)	0.61	-	-
HBP	1.40 (0.41–4.80)	0.59	-	-
Diabetes	2.22 (0.61–7.81)	0.23	-	-
Chronic renal failure	0.76 (0.13–4.19)	0.75	-	-
Active cancer	0.91 (0.16–5.17)	0.92	-	-
Antiplatelet therapy	0.66 (0.06–6.46)	0.72	-	-
Anticoagulant therapy	0.82 (0.23–2.93)	0.77	-	-
Hospitalization at diagnosis	1.23 (0.37–4.05)	0.72	-	-
Coagulation disorder				
INR	0.06 (0.76–1.46)	0.72	-	-
Hb < 7	1.31 (0.38–4.69)	0.67	-	-
MDCT Imaging data				
Retroperitoneum	4.65 (1.32–16.31)	0.016	4.08 (1.01–61.4)	0.047
Volume of hematoma (ml)	7.7 (1.86–31.92)	0.004	1.01 (1.00–1.02)	0.023
Fluid level on CT scan	0.81 (0.24–2.66)	0.727		
TAE data				
Blush at angiography	1.82 (0–∞)	0.99	-	-
Use of gelatine sponge	0.53 (0.61–1.78)	0.3	-	-
Lumbar artery	2.1 (0.61–7.22)	0.24	-	-
Number of embolized arteries	1.08 (0.23–5.04)	0.92	-	-

Abbreviations: HBP: High Blood Pressure, INR: international normalized ratio, MDCT: Multidetector Row Computed Tomography, TAE: Transarterial embolization, OR: odds ratio.

## Data Availability

The data presented in this study are available on request from the corresponding author.

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
