# Peer review of "Transarterial Embolization for Spontaneous Soft-Tissue Hematomas: Predictive Factors for Early Death"

_jpm, 2022, doi:10.3390/jpm13010015_

Round 1

Reviewer 1 Report

I am grateful to the editors of the Journal of Personalized Medicine for the opportunity to review this study. Diagnosis and treatment of spontaneous hematomas is a substantial and not fully resolved issue. In particular, spontaneous retroperitoneal hematomas in patients with venous thrombosis and pulmonary embolism are severe complications of anticoagulant therapy and are associated with a high mortality rate. In our clinic, spontaneous retroperitoneal hematomas on anticoagulant therapy occur in 0.2-0.3% of patients annually. This is a small percentage, but the treatment of such patients represents a big issue, and we remember practically each of these patients in our practice, especially when all the efforts of doctors (interventionists, abdominal surgeons, intensivists, and clinical pharmacologists) are unsuccessful and the patient dies.

In this regard, I was extremely interested to know about the possibilities and outcomes of endovascular treatment of this serious condition.

This is an interesting work performed at a high methodological level and is of undoubted interest to practicing doctors of various profiles.

At the same time, in order to improve the quality of the manuscript, I propose to make some clarifications and amendments that would improve the perception of the material by readers. I consider these comments and amendments as a minor revision. After correcting few shortcomings, the manuscript may be published in the Journal of Personalized Medicine.

Comments

1. When describing the study population, please indicate what cardiovascular diseases the patients had. Of particular interest are the percentages of patients with venous thrombosis and pulmonary embolism, prosthetic valve, cava filters, post-thrombotic syndrome, and thrombophilia/coagulopathy. Case histories should contain such data. Please clarify it.

2. Probably, it should be indicated in the text of the manuscript that all patients were elderly or senile, and their condition was severe and was accompanied with hemodynamic instability and alterations in hemostasis.

3. What were the indications to administer anticoagulant therapy? What was its regimen, and what anticoagulants did the patients receive? Please specify.

4. Did you assess the state of the blood coagulation system by only INR? Was the coagulation profile evaluated?

5. Did you discontinue anticoagulant therapy in all patients with hematomas?

6. You indicate the Clavien-Dindo classification as a tool for assessing the severity of TAE complications. Wouldn't it be better to use the SIR or CIRSE classifications, which evaluate endovascular complications?

7. How did you diagnose rebleeding after TAE? Was the diagnosis based on the trends in Hb, arterial hypotension and hematoma growth according to DUS or MSCT? In my opinion, the definition of rebleeding after TAE should be presented in the text of the manuscript.

8. You indicate that there were no complications of TAE, do I understand correctly? However, rebleeding was reported in 14.3% (Clavien-Dindo grade III, SIR class D). Isn't it a severe complication? Please clarify. I understand that the bleeding could be from another artery in the area of the hematoma or from another branch of the previously embolized artery, but formally it could be also a procedural complication. Therefore, the determination of rebleeding after TAE is necessary.

9. What complications did you consider as obvious complications of TAE? Please clarify.

10. Was the death after TAE related to the bleeding or to the heart, liver, or renal failure? If a patient died from rebleeding despite TAE, how was this interpreted? Was the endovascular treatment deemed ineffective in this case?

11. Could you please clarify timing for the 30-day mortality, i.e. did the deaths occur after 1, 7, 20 days, etc., or please indicate some range. In my opinion, this is an important point, as the cause can be different for death 1-3 days or 25-30 days after TAE.

12. You indicate that the mean volume of hematomas was 1328±578 ml. Did any of the patients undergo puncture and drainage of hematomas under ultrasound guidance? If not, why this was not done?

13. Did the hematomas cause any organ dysfunction, such as compression of the great vessels?

14. Were the platelet or plasma transfusions done for hemostasis before or after TAE? Were any other hemostatic agents used? Did this influence patient survival? This should be clarified, as the treatment of patients with retroperitoneal hematomas involves not only the use of TAE, but also a complex hemostatic therapy.

15. If the platelet or plasma transfusion or other hemostatic agents were not used, could this be a reason for the recurrence of the disease and high mortality (26.9-35.7%)?

16. In Figure 1 legend, please describe each picture as following: A [text]; B [text]; C [text].

17. In Figure 1, please indicate percentages next to the numbers.

18. I would recommend the authors to submit an angiogram of a patient with recurrent bleeding, this will certainly improve the visual perception of the material and make it more illustrative. Especially because there are only 2 figures in the manuscript.

19. In Table 4, there are 57 cases of Technical Success indicated for 56 patients. It is correct? And how to understand it? Please clarify.

20. In the Discussion section, more evidence from the literature should be presented for the treatment of spontaneous hematomas using various methods and for the benefits of TAE. These data should be provided before the discussion of your own results.

21. In the Limitations section, please indicate that the study is retrospective.

22. Conclusions in the Conclusion section should more cautious. TAE is effective in eliminating bleeding, but how effective is it in reducing mortality in patients with spontaneous hematomas? After all, the mortality during the observation period was almost 36%. As for the mortality factors - I completely agree. But the manuscript does not mention hemostatic therapy...

Once again, I thank the editors of the Journal for the opportunity to get acquainted with the manuscript, and the authors for an interesting study.

Author Response

I am grateful to the editors of the Journal of Personalized Medicine for the opportunity to review this study. Diagnosis and treatment of spontaneous hematomas is a substantial and not fully resolved issue. In particular, spontaneous retroperitoneal hematomas in patients with venous thrombosis and pulmonary embolism are severe complications of anticoagulant therapy and are associated with a high mortality rate. In our clinic, spontaneous retroperitoneal hematomas on anticoagulant therapy occur in 0.2-0.3% of patients annually. This is a small percentage, but the treatment of such patients represents a big issue, and we remember practically each of these patients in our practice, especially when all the efforts of doctors (interventionists, abdominal surgeons, intensivists, and clinical pharmacologists) are unsuccessful and the patient dies.

In this regard, I was extremely interested to know about the possibilities and outcomes of endovascular treatment of this serious condition.

This is an interesting work performed at a high methodological level and is of undoubted interest to practicing doctors of various profiles.

At the same time, in order to improve the quality of the manuscript, I propose to make some clarifications and amendments that would improve the perception of the material by readers. I consider these comments and amendments as a minor revision. After correcting few shortcomings, the manuscript may be published in the Journal of Personalized Medicine.

Comments

  1. When describing the study population, please indicate what cardiovascular diseases the patients had. Of particular interest are the percentages of patients with venous thrombosis and pulmonary embolism, prosthetic valve, cava filters, post-thrombotic syndrome, and thrombophilia/coagulopathy. Case histories should contain such data. Please clarify it.

Thank you for your suggestion to improve the manuscript. We have added patients' cardiovascular history in Table 2 of the revised manuscript.

  1. Probably, it should be indicated in the text of the manuscript that all patients were elderly or senile, and their condition was severe and was accompanied with hemodynamic instability and alterations in hemostasis.

Thank you for your suggestion. We have indicated the mean Performance status, although it is not very informative. Not all patients were senile, but a high percentage (24/56; 42.8%) of patients had a Performance Status≥2. This has been added to the revised manuscript:

“Among the population, 24/56 (42.8%) of patients had a Performance Status≥2.”

  1. What were the indications to administer anticoagulant therapy? What was its regimen, and what anticoagulants did the patients receive? Please specify.

Thank you for your suggestion to improve the manuscript. The exact regimen of anticoagulant therapy has been added in Table 2. Indications to administer anticoagulant therapy has also been added to the revised manuscript.

  1. Did you assess the state of the blood coagulation system by only INR? Was the coagulation profile evaluated?

Thank you for your comment. We had indicated the PT and platelet count in Table 2. We have also added it in the text of the manuscript for clarity.

  1. Did you discontinue anticoagulant therapy in all patients with hematomas?

Thank you for your question. All patients had discontinued therapy. Indeed, the patients in our cohort were particularly severe. In addition, we have indicated in the revised manuscript the number of patients who had anticoagulant overdose as well as the number of patients who had anticoagulant reversion.

  1. You indicate the Clavien-Dindo classification as a tool for assessing the severity of TAE complications. Wouldn't it be better to use the SIR or CIRSE classifications, which evaluate endovascular complications?

Thank you for your suggestion. We have followed your suggestion to include the number of complications according to the SIR classification.

  1. How did you diagnose rebleeding after TAE? Was the diagnosis based on the trends in Hb, arterial hypotension and hematoma growth according to DUS or MSCT? In my opinion, the definition of rebleeding after TAE should be presented in the text of the manuscript.

Thank you for your comment. The diagnosis of rebleeding was based on the drop in hemoglobin level, leading to a new CT scan showing active bleeding. All rebleeds were diagnosed by MDCT. We have added this clarification in the materials and methods section of the manuscript.

“The diagnosis of recurrent bleeding was based on the evolution of the hemoglobin level, leading to a MDCT scan.  “

  1. You indicate that there were no complications of TAE, do I understand correctly? However, rebleeding was reported in 14.3% (Clavien-Dindo grade III, SIR class D). Isn't it a severe complication? Please clarify. I understand that the bleeding could be from another artery in the area of the hematoma or from another branch of the previously embolized artery, but formally it could be also a procedural complication. Therefore, the determination of rebleeding after TAE is necessary.

According to your recommendations, we have changed the type of classification of complications in our revised manuscript. We decided not to include the number of patients with rebleeding. This is a reflection of the lack of efficacy of TAE and the severity of STTH, but not of the safety of TAE. In addition, we relied on studies in the literature (Barral et al., Dohan et al.), which do not report recurrent bleeding as a complication of TAE. A definition of complications had been added in the revised manuscript:

“Complications of TAE were defined as ischemic complication in embolized territories, non-target embolization, or hematoma at puncture site”

After careful review of the medical records, two small puncture site hematomas were reported (SIR Grade A).

  1. What complications did you consider as obvious complications of TAE? Please clarify.

Thank you for your comment. We now realize the term "obvious" is confusing. We have therefore removed it from our manuscript. We have decided to remove from the complications the occurrence of postoperative acute renal failure, the origin of which is multifactorial. However, recurrence is more a reflection of the occurrence of treatment efficacy and disease severity rather than a complication of the procedure. A definition of complications had been added in the revised manuscript:

“Complications of TAE were defined as ischemic complication in embolized territories, non-target embolization, or hematoma at puncture site”

  1. Was the death after TAE related to the bleeding or to the heart, liver, or renal failure? If a patient died from rebleeding despite TAE, how was this interpreted? Was the endovascular treatment deemed ineffective in this case?

Thank you for your comment. Most patients died from infectious, traumatic, and multifactorial complications. Indeed, the occurrence of a retroperitoneal hematoma is a factor aggravating the clinical condition of patients already hospitalized for other causes. In our study, no deaths can be directly linked to a hemorrhagic cause. All recurrent bleeding (8 patients) could be treated by a second embolization, with good technical efficacy.

  1. Could you please clarify timing for the 30-day mortality, i.e. did the deaths occur after 1, 7, 20 days, etc., or please indicate some range. In my opinion, this is an important point, as the cause can be different for death 1-3 days or 25-30 days after TAE.

Thank you for your comment to improve the quality of the manuscript. We have added a Kaplan-Meier curve (Figure 4). We have also added in Table 3 the day-3 mortality (8.9%).

  1. You indicate that the mean volume of hematomas was 1328±578 ml. Did any of the patients undergo puncture and drainage of hematomas under ultrasound guidance? If not, why this was not done?

Thank you for your question. No surgical management or radiological drainage of hematomas were performed.  The risk of surgical treatment is to achieve a rapid decompression of the hematoma and thus cause a recurrence of bleeding. The following sentence has been added to the manuscript:

“No surgical management or radiological drainage of the hematoma were performed.”

  1. Did the hematomas cause any organ dysfunction, such as compression of the great vessels?

Thank you for your suggestion. We observed in our cohort a partial renal ischemia by compression of the renal artery. The following sentence was added to the revised manuscript:

“One patient with a retroperitoneal hematoma had partial ischemia of the kidney on arterial plication by extrinsic compression of the hematoma.”

  1. Were the platelet or plasma transfusions done for hemostasis before or after TAE? Were any other hemostatic agents used? Did this influence patient survival? This should be clarified, as the treatment of patients with retroperitoneal hematomas involves not only the use of TAE, but also a complex hemostatic therapy.

Thank you for your comment which allows us to further improve the manuscript. Indeed, plasma or platelet transfusion were reported in patients. We report transfusions performed within 24 hours before and 48 hours after the procedure. Indeed, it is not possible to report precisely how many transfusions were performed before and after. In addition, some transfusions are usually performed during the procedure.

 We repeated the statistical analysis which did not show the influence of plasma or platelet transfusion rate on early mortality.

  1. If the platelet or plasma transfusion or other hemostatic agents were not used, could this be a reason for the recurrence of the disease and high mortality (26.9-35.7%)?

Thank you for your comment. We apologize for not having indicated in the original manuscript the rate of patients with platelet and plasma transfusion. We have added it in the revised manuscript.

  1. In Figure 1 legend, please describe each picture as following: A [text]; B [text]; C [text].

Thank you for your suggestion. We have modified Figure 1.

  1. In Figure 1, please indicate percentages next to the numbers.

Thank you for your suggestion. We have modified Figure 1.

  1. I would recommend the authors to submit an angiogram of a patient with recurrent bleeding, this will certainly improve the visual perception of the material and make it more illustrative. Especially because there are only 2 figures in the manuscript.

Thank you for your comment. We have added a figure showing a patient with recurrent bleeding after TAE. Here is the figure added to the revised manuscript:

Figure 3: Back pain in a 45-year-old patient hospitalized for Sars-cov2 infection and pulmonary embolism. (A) The MDCT scan showed active bleeding from a large left retroperitoneal hematoma. (B) Angiography confirmed active unifocal bleeding from the left L5 lumbar artery, treated with gelatine sponge(C). On day 1, the patient presented a persistent blood loss. The MDCT scan showed a persistent hematoma with active bleeding (arrow). (D) Angiography showed a new multifocal active bleeding treated using 1 coil.    

  1. In Table 4, there are 57 cases of Technical Success indicated for 56 patients. It is correct? And how to understand it? Please clarify.

We apologize for this confusing indication in Table 4. Only one patient had a technical failure as shown in Figure 2, so this patient was not included in the analysis because he did not benefit from TAE. We removed this misleading line from the table.

  1. In the Discussion section, more evidence from the literature should be presented for the treatment of spontaneous hematomas using various methods and for the benefits of TAE. These data should be provided before the discussion of your own results.

Thank you for your suggestion. We have added a sentence in the "Discussion" section to report the results of the literature.

  1. In the Limitations section, please indicate that the study is retrospective.

Thank you for your comment. We have added the following sentence to the revised manuscript:

“It is a single-centre retrospective study”

  1. Conclusions in the Conclusion section should more cautious. TAE is effective in eliminating bleeding, but how effective is it in reducing mortality in patients with spontaneous hematomas? After all, the mortality during the observation period was almost 36%. As for the mortality factors - I completely agree. But the manuscript does not mention hemostatic therapy...

Thank you for your comment. You are indeed right. Embolization is effective on the risk of bleeding but no study has really shown its interest in reducing mortality. We have added the following sentence in the revised manuscript:

“Nevertheless, early death remains high.”

Moreover, the description of the patient's main symptom has been added in the revised manuscript. It is interesting to note that 26/56(46.4%) patients had no clinical symptomatology apart from hypotension and/or drop in hemoglobin.

We have added the number of patients who had an anticoagulant overdosage. Definitions of hemoglobin drop and hemodynamic instability have been added.

Once again, I thank the editors of the Journal for the opportunity to get acquainted with the manuscript, and the authors for an interesting study.

Reviewer 2 Report

The objective of the article entitled “Trans arterial embolization for spontaneous soft-tissue hematomas: predictive factors for early death” was to identify predictive factors of risk of death in patients with retroperitoneal hematoma. It would have been of a major interest unfortunately the article is difficult to read since it is written in a poor language and not rigorously structured.

-          The table 1 is presented inadequately indicating patient characteristics as variable in percentage

The authors included only patients who were treated by embolization and they did not compare to a control group receiving another treatment. Dohan A. et al published in 2015 an article of the objective was to evaluate the effectiveness and the safety of embolization; this study was a prospective study. They concluded that overdosage of anticoagulant was found in 33% of the patients and that embolization reduced the need of blood transfusion. Touma L. et al (Dohan A. last author) published in 2019 a review which demonstrated the efficacy and safety of TAE for the management of SSTH (63 studies, 267 patients).

The results analyzed in this article did not provide a great help to the MD taking care of this hemorrhagic complication and did not clear what is the appropriate anticoagulation and antithrombotic treatment. It would have been also of interest to know what were the symptoms and the management of the patients beside blood transfusion.

It is surprising that the age was not a risk factor, as well as the intensity of bleeding based on hemoglobin blood level. Only the size of hematoma appeared as a risk factor.

The authors detailed in their conclusion what is the weakness of the article and may answer themselves the points before submitting the article which at least needs major revisions.

This article is a fade copy of the two articles published by Dohan group and did not provide new information.

Author Response

The objective of the article entitled “Trans arterial embolization for spontaneous soft-tissue hematomas: predictive factors for early death” was to identify predictive factors of risk of death in patients with retroperitoneal hematoma. It would have been of a major interest unfortunately the article is difficult to read since it is written in a poor language and not rigorously structured.

-          The table 1 is presented inadequately indicating patient characteristics as variable in percentage

The authors included only patients who were treated by embolization and they did not compare to a control group receiving another treatment. Dohan A. et al published in 2015 an article of the objective was to evaluate the effectiveness and the safety of embolization; this study was a prospective study. They concluded that overdosage of anticoagulant was found in 33% of the patients and that embolization reduced the need of blood transfusion. Touma L. et al (Dohan A. last author) published in 2019 a review which demonstrated the efficacy and safety of TAE for the management of SSTH (63 studies, 267 patients).

The results analyzed in this article did not provide a great help to the MD taking care of this hemorrhagic complication and did not clear what is the appropriate anticoagulation and antithrombotic treatment. It would have been also of interest to know what were the symptoms and the management of the patients beside blood transfusion.

It is surprising that the age was not a risk factor, as well as the intensity of bleeding based on hemoglobin blood level. Only the size of hematoma appeared as a risk factor.

The authors detailed in their conclusion what is the weakness of the article and may answer themselves the points before submitting the article which at least needs major revisions.

This article is a fade  copy of the two articles published by Dohan group and did not provide new information.

Thank you for taking the time to review the manuscript. We have made substantial improvements to the manuscript, both in form and content. Thank you for allowing us to improve our manuscript.

You referred to the two articles by the Dohan team. This team is indeed a pioneer in the description of SSTH TAE. The first article published is a retrospective paper (as described in the limitations section of their study) on safety and efficacy. The second study is a review, which shows the small number of studies and cases reported in the literature. This review focuses on the efficacy and safety of SSTH embolization, but not on the predictive factors of mortality after SSTH TAE. Our study is second study, following the one from Barral et al., that investigated the predictive factors of survival after SSTH TAE. It concludes that the same predictive factors influence survival: retroperitoneal location and volume of the hematoma. Thus, it puts the diagnostic radiologist at the centre of the management. Our study does not aim to give advice and indications on the management of anticoagulant treatment.

We did not expect that age and hemoglobin level would not be predictive of mortality. However, this is not totally surprising. Indeed, the study by Barral et al. also showed that age and hemoglobin level were not predictive of mortality after SSTH TAE. In this respect, our article is in line with the literature. 

We have made structural improvements of the manuscript:

1.The antithrombotic therapy has been described in Table 2.

2.The indication for antithrombotic therapy has been added.

3.A Kaplan-Meier curve to report the survival rate in the first 100 days has been added in the revised manuscript.

Figure 4: Kaplan-Meier curve of overall survival during the first 100 days after TAE

4.A figure illustrating a recurrent bleeding at day 1 has been added in the revised manuscript.

Figure 3: Back pain in a 45-year-old patient hospitalized for Sars-cov2 infection and pulmonary embolism. (A) The MDCT scan showed active bleeding from a large left retroperitoneal hematoma. (B) Angiography confirmed active unifocal bleeding from the left L5 lumbar artery, treated with gelatine sponge(C). On day 1, the patient presented a persistent blood loss. The MDCT scan showed a persistent hematoma with active bleeding (arrow). (D) Angiography showed a new multifocal active bleeding treated using 1 coil.    

5.We had another Native English speaker proofread the document for clarity and have provided a proofreading certificate.

6.The description of the patient's main symptoms has been added, as you requested, in the revised manuscript. It is interesting to note that 26/56(46.4%) patients had no clinical symptomatology apart from hypotension and/or drop in hemoglobin.

7.The number of patients receiving platelet and plasma transfusions had been added.

8.The number of patients who received reversal therapy has been specified. The type of reversal therapy had been detailed in table 2. We have also indicated the rate of anticoagulant overdosage.

9.We specified the number of occurrences of extrinsic organ compression by the hematoma. One extrinsic compression of renal artery has been reported.

10.We specified in the revised manuscript that no surgical decompression maneuvers were performed.

  1. We have reorganized the Tables and the materials and methods section for clarity: “patient characteristics”, “imaging and procedure data”, “outcomes and predictive factors”.

Thank you for reviewing this manuscript. Your comments have helped us to improve the overall quality.

Round 2

Reviewer 2 Report

The work is important but we are disappointed by the fact that the authors take only into account a part of the points raised by the reviewers. It is not smart to submit an amended version in a form of article first draft. The use of red color is normally limited. In the fraction of the text which is readable the authors added new errors.

The rule for drugs is to cite the international trade name; for example, Plavix is Clopidogrel, Heparin is a therapeutical family.

The authors made some modifications of the article but they did not provide individual values of the patients. It is not possible to check the authors’ conclusion with only the mean value ± SD. The authors wrote that the article was for radiologists, if it is their objective , they possibly select a more specialized journal.

The authors mentioned figure 1 page 13 where is the figure 1? Did the authors made an error about patients’ outcomes, are they detailed figure 2 or table 4?

 Beside the numerous corrections of English, the article is not easy to read and could not be accepted in the present form for publication.

Author Response

The work is important but we are disappointed by the fact that the authors take only into account a part of the points raised by the reviewers. It is not smart to submit an amended version in a form of article first draft. The use of red color is normally limited. In the fraction of the text which is readable the authors added new errors.

The rule for drugs is to cite the international trade name; for example, Plavix is Clopidogrel, Heparin is a therapeutical family.

The authors made some modifications of the article but they did not provide individual values of the patients. It is not possible to check the authors’ conclusion with only the mean value ± SD. The authors wrote that the article was for radiologists, if it is their objective , they possibly select a more specialized journal.

The authors mentioned figure 1 page 13 where is the figure 1? Did the authors made an error about patients’ outcomes, are they detailed figure 2 or table 4?

 Beside the numerous corrections of English, the article is not easy to read and could not be accepted in the present form for publication.

Thank you for your comments and suggestions. We have changed means and standard deviations for median and interquartile ranges. In addition, we replaced the term "Plavix" with "clopidogrel" and specified the type of heparin used in Table 2.

Table 2: Patient biological data

Variables

56

Antithrombotic therapy n.(%)

51

Antiplatelet  

5(8.9)

Clopidogrel

2

Aspirin

3

Anticoagulant 

39(69.6)

LMWH

7

UFH

14

VKA

12

Apixaban

3

Rivaroxaban

2

Antiplatelet and Anticoagulation  

7(12.5)

VKA + Aspirin

1

UFH + Aspirin

6

No antithrombotic therapy

5(8.9)

Discontinuation of anticoagulant therapy n. (%)

51(100)

Overdosage n. (%)

13(25.5)

Reversion of anticoagulant therapy n. (%)

13(25.5)

Prothrombin complex concentrates

10(19.6)

K Vitamin

9(17.6)

Tranexamic Acid

5(9.8)

Biology, median [Q25-75] 

INR

1.45[1-2.7]

PT(%)

65[8-75]

Platelet (G/L)

180[38-255]

Hemoglobin (g/dl)

7.6[4.4-8.2]

Drop of Hemoblogin (g/dl)

3.45[0.9-4.9]

RBC Transfusion n. (%)

41(71.9)

Number of RBC, median [Q25-75] 

4[1-7.5]

Fresh frozen plasma transfusion n. (%)

25(44.6)

Number of fresh frozen plasma units, median [Q25-75] 

3[1-6]

Platelet transfusion n. (%)

7(12.5)

Number of platelet units, median [Q25-75] 

2[2-3]

Abbreviations: INR: international normalized ratio, PT: prothrombin time test, UFH: Unfractioned heparin, LMWH: Low-molecular-weight Heparin

Figure 2 and Table 4 show information on patients’ outcomes. Figure 2 is the flow chart of patient outcomes. Table 4 also details patients’ outcomes.

Figure 1 is in the revised manuscript that you have sent us. We provide it below:

Figure 1: Isolated blood loss in a 74-year-old patient 2 days after cardiac surgery. (A)The MDCT scan showed active bleeding from a large left retroperitoneal hematoma. (B) Angiography confirmed active multifocal bleeding from the left L5 lumbar artery. (C) After embolization with NBCA, the control showed no opacification of the distal branches of the left L5 lumbar artery.

Our article is not intended specifically for the radiological community. However, we have submitted this article for a special issue focused on interventional radiology and embolization. We provide very detailed clinical information that will be useful to all clinicians who manage these patients. In addition, our study is only the 2nd to focus on prognostic factors after embolization of SSTH. What do you mean by “individual values”? We do not understand the meaning of your comment. We have provided additional information on symptoms, individual patient management, which we express using percentages and medians.

You mention that the article is not easy to read: can you be more precise? We are surprised by your comment because we respect the organization of articles published in the field of interventional radiology. In fact, the paragraphs in our article are organized like those of the Dohan’s team. In addition, we would like to remind you that we have had the English version of the article proofread.